Evaluation of optic nerve sheath diameter in acute stroke: pre- and post-thrombolytic assessment

Sivas Erdem 1 doktorerdem66@gmail.com
http://orcid.org/0000-0001-6821-9031 Colak Nese 2
Bayram Basak 3
Simsek Muhammet Kursat 4
Karabay Nuri 5
Ozturk Vesile 6
1 Department of Emergency Medicine, Mardin Public Hospital , Mardin , Turkey
2 Department of Emergency Medicine, University of Health Sciences Turkey , Izmir , Turkey
3 Department of Emergency Medicine, Kocaeli City Hospital , Kocaeli , Turkey
4 Department of Radiology, Manisa Provincial Health Directorate Merkezefendi Public Hospital , Manisa , Turkey
5 Department of Radiology, Dokuz Eylul University, Faculty of Medicine , Izmir , Turkey
6 Department of Neurology, Dokuz Eylul University, Faculty of Medicine , Izmir , Turkey
Kashoo Faizan
Electronic publication date: 2025 Mar 31
Publication date: 2025
Volume: 13
Electronic Location ID: e19197
Received 2024 Sep 19; Accepted 2025 Feb 28
Copyright: © 2025 Sivas et al.
Copyright year: 2025
Copyright holder: Sivas et al.
License: This is an open access article distributed under the terms of the Creative Commons Attribution License, which permits unrestricted use, distribution, reproduction and adaptation in any medium and for any purpose provided that it is properly attributed. For attribution, the original author(s), title, publication source (PeerJ) and either DOI or URL of the article must be cited.
License URL: https://creativecommons.org/licenses/by/4.0/

Keywords: Optic nerve sheath diameter, Acute ischemic stroke, Thrombolytic treatment, Stroke complication, Mortality

Funding: The authors received no funding for this work.

==============================
Background

Intracranial pressure increases due to ischemic infarction caused by stroke. This study aimed to evaluate the pre-thrombolytic and post-thrombolytic optic nerve sheath diameter (ONSD) measurements in predicting clinical outcomes and complications for stroke patients.

Methods

ONSD was measured on computed tomography (CT) scans. The average ONSD (aONSD) was calculated from the right and left eyes. Pre-thrombolytic (ONSD-0) and post-thrombolytic (ONSD-24) values were compared according to right vs left eye, stroke-affected side of the brain, presence of complications, and mortality.

Results

Ninety-three patients were enrolled; 52.7% were female, and the mean age of all participants was 76 years. The aONSD-24 values were higher than the aONSD-0 values (5.5 ± 0.7 mm and 5.3 ± 0.7 mm, respectively, p < 0.001). There was no significant difference between right and left measurements (right ONSD-0 5.3 mm vs. left ONSD-0 5.3 mm, p = 0.257; right ONSD-24 5.6 mm vs. left ONSD-24 5.5 mm, p = 0.146; and ∆right ONSD 0.23 mm vs. ∆left ONSD 0.22, p = 0.717) and between the stroke-affected side and non-stroke-affected side measurements (stroke-affected ONSD-0 5.2 mm vs. non-stroke-affected ONSD-0 5.2 mm, p = 0.292; stroke-affected ONSD-24 5.5 mm vs. non-stroke-affected ONSD-24 5.4 mm, p = 0.124; and ∆stroke-affected ONSD 0.23 mm vs. non-∆stroke-affected ONSD 0.23 mm, p = 0.569). Intracranial complications occurred in 14 (15%) patients. There was no difference in ONSD values between patients with and without complications (p = 0.338 for aONSD-0, p = 0.216 for aONSD-24, and p = 0.902 for ∆a ONSD). There was no significant difference between the aONSD-0 and aONSD-24 values of surviving and non-surviving patients (aONSD-0: 5.3 ± 0.7 vs. 5.0 ± 0.5, p = 0.345; aONSD-24: 5.5 ± 0.7 vs. 5.3 ± 0.4, p = 0.522; and p = 0.386 for ∆ aONSD).

Conclusions

ONSD values on 24-h brain CT scans were higher than admission values in acute stroke patients receiving thrombolytic therapy, irrespective of the right or left side, stroke-affected side, presence of complications, and mortality. However, ONSD is not a sufficient parameter for predicting complications and death.

Introduction

In ischemic stroke, edema developing in infarcted tissue can lead to increased intracranial pressure (ICP), ranging from asymptomatic progression to severe complications, including death, depending on the severity of the stroke and on patient factors (Berrouschot et al., 1998; Gu et al., 2022). Cytotoxic edema occurs rapidly after ischemic stroke, followed by ionic, vasogenic, and mixed edema, which typically peaks within 3–5 days of stroke onset (Dostovic et al., 2016).

Optic nerve sheath diameter (ONSD) is a non-invasive surrogate for ICP measurement, with strong evidence supporting its correlation with elevated ICP (Lee et al., 2018; Raffiz & Abdullah, 2017; Tayal et al., 2007). As the optic nerve sheath is continuous with the dura mater and subarachnoid space, increased ICP causes measurable dilation. Studies indicate that the sensitivity and specificity of ONSD for detecting elevated ICP can vary depending on the population, methodology, and cutoff values used. Studies have proposed ONSD cut-off values, commonly ranging between 5 and 5.8 mm (Tayal et al., 2007; Geeraerts et al., 2007; Goel et al., 2008). Using repeated measurements to detect dynamic changes in ONSD over time, rather than relying on a single cut-off value, may provide a stronger basis for identifying ICP changes.

In acute stroke, cranial computed tomography (CCT) is routinely performed both before and 24 h after thrombolytic therapy. ONSD measurement on CCT images is a practical, non-invasive diagnostic tool in emergency settings, requiring no additional imaging beyond routine stroke evaluation. This study aimed to compare pre- and post-thrombolytic ONSD measurements on CCT images and investigate the relationship of these measurements with clinical improvement and complication development. This protocol provides an opportunity to evaluate ONSD changes and their association with clinical outcomes.

Materials and Methods

In this observational, cross-sectional, and retrospective study, all consecutive patients presenting with acute stroke and receiving thrombolytic therapy at the emergency department of Dokuz Eylul University Hospital between 01/01/2015 and 01/06/2018 were included. The following patients were excluded from the study: those in whom ONSD could not be measured in both eyes, those with cranial-orbital trauma, those who had previously been diagnosed with elevated ICP and additional pathological diseases, those with a history of cranial surgery, and those without a 24-h control CCT.

This study was approved by the Clinical Studies Ethics Committee of Dokuz Eylul University Faculty of Medicine (decision number: 2018/24-14 date: 04.10.2018) and a waiver of informed consent was granted due to the retrospective nature of the study and the use of de-identified data. Patient information was accessed via the Hospital Information Management System (HIMS). The following patient data were recorded: patient complaints on admission, age, gender, haemodynamic parameters, comorbidities, and post-thrombolytic intracranial complications (intracranial hemorrhage, cerebral edema).

ONSD measurements were performed by a neuroradiology faculty member on CCT images obtained using a Toshiba Aquilion Prime device. Measurements were taken from the axial section 3 mm posterior to the eyeball (Fig. 1). The ONSD values at admission (ONSD-0) and 24 h after thrombolysis (ONSD-24) were recorded. The average of the right and left eye measurements was calculated as the average ONSD (aONSD). Additionally, the ONSD was categorized as either a right ONSD or a left ONSD, and as either a stroke-affected ONSD (the side corresponding to the ischemic lesion) or a non-stroke-affected ONSD (the side without ischemic involvement). To assess changes over time, the difference between the ONSD-24 value and the ONSD-0 value was calculated and termed ∆ONSD. Finally, ONSD values were compared with respect to the presence of complications and mortality.

Figure 1 ONSD measurement.

Statistical analysis

The SPSS 29.0 (IBM Corporation, Armonk, NY, USA) program was used to analyze the data. Categorical variables were written as numbers and percentages, and the Pearson chi-square or Fisher exact test was used to compare these data. Normality analyses were evaluated with the Kolmogorov-Smirnov test, and standard deviation values were reported. The independent samples t-test was used in the independent groups (aONSD values for patients with complications vs. without complications, and surviving vs. non-surviving patients), and the paired samples t-test was used in the dependent groups (ONSD-0 vs. ONSD-24, right ONSD vs. left ONSD, and stroke-affected ONSD vs. non-stroke-affected ONSD). P-value < 0.05 was considered significant.

Results

A total of 95 patients who were admitted to the emergency department with acute stroke and received thrombolytic therapy between 1 January 2015 and 1 June 2018 were included in the study. Two patients were excluded because they lacked a 24-h CCT. The mean age of included patients was 74 ± 12 years, and 49 out of 93 included patients (52.7%) were female. The majority of patients had at least one comorbid condition, with hypertension being the most common. Demographic and clinical characteristics of the patients are shown in Table 1.

Table 1 Demographics, vital signs, and comorbidities of the patients.

Demographics	Mean ± SD/n	Min-max/%	
Age (years)	74 ± 12	37–94	
Gender (Female/Male)	49/44	53%/47%	
Vital signs			
Systolic blood pressure (mmHg)	158 ± 28	97–223	
Diastolic blood pressure (mmHg)	91 ± 16	57–136	
Pulse (/min)	82 ± 18	53–143	
Respiratory rate (/min)	21 ± 2	16–28	
Oxygen saturation (%)	96 ± 2	91–99	
Body temperature (°C)	26.3 ± 4.5	35.1–38.1	
Comorbidities			
Hypertension	65	69.9%	
Diabetes mellitus	25	26.9%	
Coronary artery disease	16	17.2%	
Heart failure	14	15.1%	
İschemic stroke event	11	11.8%	
Arrhythmia	7	7.5%	
Hyperlipidemia	3	3.2%	
Others	27	29%	

The anatomical location of the stroke was determined in 83/93 patients, with middle cerebral artery (MCA) infarction being the most common, followed by posterior cerebral artery (PCA) infarction (Table 2).

Table 2 Stroke locations of the patients.

Anatomic location	Right n (%)	Left n (%)	
MCA	19 (20.4)	30 (32.3)	
PCA	6 (6.5)	5 (5.4)	
ACA	6 (6.5)	0 (0.0)	
Thalamic	4 (4.3)	4 (4.3)	
Lacunar	2 (2.2)	1 (1.1)	
Cerebellar	0 (0.0)	2 (2.2)	
Brain stem	4 (4.3)		
Unknown	10 (10.8)		
Total	93 (100)		

Table 3 shows the aONSD results for the right and left sides. The ONSD-24 values were found to be statistically significantly higher than the ONSD-0 values between the right eye and the left eye and between the stroke-affected side and the non-stroke-affected side. Conversely, there were no significant differences between the ONSD-0 values (p = 0.257), ONSD-24 values (p = 0.146), and ∆ONSD (p = 0.569) in the right and left side measurements. The correlation coefficient between the right ONSD and left ONSD was calculated to be 0.809 for ONSD-0 and 0.760 for ONSD-24. The average of the right and left eye values were then calculated, and the aONSD-24 values were found to be 5.5 ± 0.7 mm and the aONSD-0 values were 5.3 ± 0.7 mm (p < 0.001).

Table 3 ONSD-0, ONSD-24 and ∆ONSD values of the patients.

Time	ONSD-0 (mm)	ONSD-24 (mm)	∆ONSD (mm)	p-value	
Mean ± SD (min-max)	Mean ± SD (min-max)	Mean ± SD (min-max)		
Right eye (n = 93)	5.3 ± 0.8 (3.6–8.4)	5.6 ± 0.7 (4.0–8.0)	0.23 ± 0.38 ([−]0.7–1.7)	<0.001#	
Left eye (n = 93)	5.3 ± 0.7 (3.8–7.6)	5.5 ± 0.7 (4.1–7.5)	0.22 ± 0.37 ([−]0.7–1.7)	<0.001#	
p-value	0.257#	0.146#	0.717#		
Stroke-affected side (n = 65)	5.2 ± 0.7 (4.0–8.4)	5.5 ± 0.8 (4.1–8.0)	0.23 ± 0.36 ([−]0.7–1.0)	<0.001#	
Non-stroke-affected side (n = 65)	5.2 ± 0.8 (3.6–7.6)	5.4 ± 0.7 (4.0–7.5)	0.23 ± 0.29 ([−]0.7–1.3)	<0.001#	
p-value	0.292#	0.124#	0.569#		
	a-ONSD-0 (mm)	a-ONSD-24 (mm)	∆a-ONSD (mm)		
Mean ± SD (min-max)	Mean ± SD (min-max)	Mean ± SD (min-max)		
Without complication (n = 79)	5.3 ± 0.8 (4.0–8.0)	5.6 ± 0.7 (4.3–7.8)	0.23 ± 0.31 ([−]0.7–1.1)	<0.001#	
With complication (n = 14)	5.2 ± 0.4 (4.5–5.9)	5.4 ± 0.3 (4.8–5.9)	0.22 ± 0.17 ([−]0.15–0.45)	<0.001#	
p-value	0.338¥	0.216¥	0.902¥		
Surviving (n = 88)	5.3 ± 0.7(4.0–8.0)	5.5 ± 0.7 (4.3–7.8)	0.22 ± 0.30 ([−]0.7–1.1)	<0.001#	
Non-surviving (n = 5)	5.0 ± 0.5(4.5–5.6)	5.3 ± 0.4 (4.8–5.8)	0.34 ± 0.13 ([−]0.15–0.45)	0.042#	
p-value	0.345¥	0.522¥	0.386¥		
Notes:

# Paired sample t-test.

¥ Independent sample t-test.

In 65 patients in whom the stroke-affected side of the brain could be assessed, the stroke-affected ONSD-24 values were higher than the stroke-affected ONSD-0 values (p < 0.001). The non-stroke-affected ONSD-24 values were also higher than the non-stroke-affected ONSD-0 values (p < 0.001). There were no differences between the stroke-affected and non-stroke-affected ONSD-0, ONSD-24, and ∆aONSD values (ONSD-0 p = 0.292, ONSD-24 p = 0.124, and ∆aONSD p = 0.902). The correlation coefficient between the stroke-affected and non-stroke-affected sides was calculated to be 0.809 for aONSD-0 and 0.749 for aONSD-24 (Table 3).

Fourteen (15%) patients who received thrombolytics developed intracranial complications. Eight patients developed intracranial hemorrhage, two experienced cerebral edema, and four developed intracranial hemorrhage with cerebral edema. The aONSD-24 values were higher than the aONSD-0 values in both patients with and without complications (both p < 0.001). There was no significant difference between patients with and without complications according to aONSD-0, aONSD-24, and ∆aONSD values (aONSD-0 p = 0.338, ONSD-24 p = 0.216, and ∆aONSD p = 0.386; Table 3).

All included patients survived the first 24 h after admission to the emergency department, although five patients (5.4%) died between 24 h and day seven. There were no significant differences between the aONSD-0, aONSD-24, and ∆aONSD values of the surviving and non-surviving patients (respectively aONSD-0: 5.3 ± 0.7 vs. 5.0 ± 0.5, p = 0.345; aONSD-24: 5.5 ± 0.7 vs. 5.3 ± 0.4, p = 0.522; and ∆aONSD: 0.22 ± 0.30 vs. 0.34 ± 0.13, p = 0.386).

The ONSD values according to patient complications and the stroke-affected areas of the patients with complications are shown in Table 4.

Table 4 Affected region and ONSD values of patients with complications.

Patient number	Age	Sex	Affected
area	Complication	ONSD-0 (mm)	ONSD-24 (mm)	
Stroke-affected side	Non-stroke-affected side	Stroke-affected side	Non-stroke-affected side	
1	72	Male	Left MCA	Brain edema and shift 17 mm	5.7	5.0	6.1	5.5	
2	76	Male	Right ACA and MCA	Intracerebral hemorrhage	5.8	6.0	6.0	5.8	
3	93	Female	Left MCA	Intracerebral hemorrhage, brain edema 18 mm shift	5.4	5.7	5.8	5.6	
4	94	Female	Right ACA and MCA	Intracerebral hemorrhage and brain edema	4.9	4.9	5.4	5.3	
5	57	Female	Right MCA	Intracerebral hemorrhage	6.3	5.2	6.2	5.0	
6	71	Male	Left MCA	Intracerebral hemorrhage, brain edema 17 mm shift	4.0	4.9	4.6	4.9	
7	67	Male	Left PCA	Intracerebral hemorrhage	4.8	5.1	5.1	5.4	
8	45	Male	Right ACA and MCA	Intracerebral hemorrhage	4.8	4.7	5.7	4.6	
9	88	Female	Left MCA	Intracerebral hemorrhage	4.6	5.0	4.7	5.6	
10	77	Male	Left MCA	Intracerebral hemorrhage, brain edema	5.3	5.0	5.2	5.6	
11	62	Female	Left MCA	Brain edema 13 mm shift	5.4	5.6	5.7	5.6	
12	78	Female	Left MCA	Brain edema 17 mm shift	5.1	4.9	5.6	4.7	

Discussion

The present study investigated the pre- and post-thrombolytic ONSD values in patients with ischemic stroke who received thrombolytic therapy. There are a limited number of previous studies on the measurement of ONSD in ischemic or hemorrhagic cerebrovascular disease (Tayal et al., 2007; Geeraerts et al., 2007; Goel et al., 2008; Yesilaras et al., 2017; Gökcen et al., 2017). While most studies have utilized ultrasound for ONSD measurements due to its repeatable, non-invasive, and bedside applicability, ultrasound is inherently operator-dependent, leading to variability based on clinician expertise. CT, in contrast, provides more objective and consistent results, minimizing inter-operator variability. The disadvantages of CT include radiation exposure and the need to transport the patient to the imaging suite, which might not always be feasible in unstable patients. CT is routinely performed in acute stroke management, making it a practical choice for this study’s retrospective design.

Optic nerve sheath diameter measurements (ONSD) are a widely studied tool for estimating ICP, which is an important cause of secondary cerebral injury in stroke patients. In a randomized controlled study by Gökcen et al. (2017) of 191 patients, a significant increase in bilateral ONSD was observed in ischemic stroke patients compared to the control group. While single ONSD cutoff values, typically ranging from 5.0 mm to 6.4 mm, are commonly used to detect elevated ICP, these values can be affected by individual anatomical and physiological differences (Tayal et al., 2007; Geeraerts et al., 2007; Goel et al., 2008; Bansal et al., 2024). Studies show that repeated measurements of ONSD under different conditions or at different points in time may provide more comprehensive data about trends in ICP than relying on a single cutoff value, which could be affected by individual anatomical or physiological differences (Bansal et al., 2024; Chen et al., 2023). In pediatric populations, for example, sequential ONSD measurements have shown significant differences between groups with raised ICP and normal ICP, underscoring the importance of tracking trends rather than relying on isolated measurements (Chen et al., 2023). Thus, repeated ONSD measurements may offer a more robust method for detecting ICP changes, accounting for the dynamic nature of ICP fluctuations and inter-individual variability.

For the present study, patients were screened according to their 0 and 24-h ONSD measurements and screened separately for each of the following: the stroke-affected regions, the non-stroke-affected regions, and the development of intracranial complications. Although there are existing studies investigating the relationship between ONSD measurement and ICP and stroke, there are no studies differentiating between ONSD measurements of the stroke-affected side and non-stroke-affected side or between patients who developed complications and those who did not. According to the results of the current study, a significant difference was observed between the ONSD measurements of both the stroke-affected and non-stroke-affected sides at 0 and 24 h. While CCT findings and ICP are normal in stroke patients in the early period, ICP and ICP findings due to cytotoxic edema become more pronounced in the following hours. Therefore, an increase in ONSD with ICP and edema findings, as observed in the present study, is an expected result.

In a study by Bekerman et al. (2016) examining cases with ICP, an increase in ONSD was observed in 94.3% of the patients, and the ONSD was found to be 6.3 ± 0.9 mm in the left eye and 6.2 ± 1.2 mm in the right eye. The current study examined the relationship between the development of intracranial complications and ONSD, regardless of the stroke-affected area. There were no significant differences between the groups with and without complications in the right- and left-sided characteristics of the right ONSD-0, left ONSD-0, right ONSD-24, and left ONSD-24 measurements. When evaluated according to the characteristics of the stroke-affected and non-stroke-affected regions, a significant difference was found between ONSD-0 and ONSD-24 values in the region affected by the stroke in the patients who developed complications, but there was no significant difference between ONSD-0 and ONSD-24 values in the non-stroke-affected region. In patients who did not develop complications, a significant difference was observed between ONSD-0 and ONSD-24 values in both the stroke-affected and non-stroke-affected areas. Theoretically, ICP is expected to be higher in patients with intracranial complications due to the development of intracranial hemorrhage, edema, and shift (Schwab et al., 1996). This increased ICP would lead to an increase in ONSD. This expected result was seen in the present study, where a significant increase in ONSD was observed in patients with complications compared to patients without complications. Modified Rankin scale (mRS) scores and ONSD changes were evaluated, and no significant relationship with mRS was observed.

Limitations

The main limitations of the current study are that there is no international standard measurement method for the measurement of ONSD and that ONSD measurements were performed by a single radiologist. Inter-individual variability in ONSD measurements should be considered. Complications that developed in the patients were evaluated, however, because there is no objective parameter to measure the response to thrombolytic therapy, the success of thrombolytic therapy and its effect on ONSD could not be evaluated separately.

This retrospective observational study relied on routine clinical imaging protocols for acute stroke patients, with non-contrast CCT performed before thrombolysis and 24 h after treatment, according to guidelines. Consequently, ONSD measurements were limited to these two time points. A prospective study with more frequent imaging intervals may provide a better understanding of the temporal evolution of ONSD changes.

Conclusions

The present study found that ONSD values on 24-h brain CT scans were higher than admission values in acute stroke patients receiving thrombolytic therapy, irrespective of the right or left side, stroke-affected side, presence of complications, and mortality. However, ONSD is not a sufficient parameter for predicting complications and death.

Supplemental Information

Supplemental Information 1 Study data.

Data were recorded using the SPSS 29.0 program (IBM Corporation, Armonk, New York, United States).

Additional Information and Declarations

Competing Interests

The authors declare that they have no competing interests.

Author Contributions

Erdem Sivas conceived and designed the experiments, performed the experiments, analyzed the data, prepared figures and/or tables, authored or reviewed drafts of the article, and approved the final draft.

Nese Colak conceived and designed the experiments, performed the experiments, analyzed the data, prepared figures and/or tables, authored or reviewed drafts of the article, and approved the final draft.

Basak Bayram conceived and designed the experiments, performed the experiments, analyzed the data, prepared figures and/or tables, authored or reviewed drafts of the article, and approved the final draft.

Muhammet Kursat Simsek performed the experiments, authored or reviewed drafts of the article, and approved the final draft.

Nuri Karabay conceived and designed the experiments, analyzed the data, authored or reviewed drafts of the article, and approved the final draft.

Vesile Ozturk conceived and designed the experiments, analyzed the data, authored or reviewed drafts of the article, and approved the final draft.

Human Ethics

The following information was supplied relating to ethical approvals (i.e., approving body and any reference numbers):

The study was started after obtaining the permission of the Dokuz Eylül University Faculty of Medicine Clinical Studies Ethics Committee (Decision no: 2018/24-14 Date: 04.10.2018).

Data Availability

The following information was supplied regarding data availability:

The raw measurements are available in the Supplemental File.

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
