# Peer review of "Evaluation of optic nerve sheath diameter in acute stroke: pre- and post-thrombolytic assessment"

_PeerJ, doi:10.7717/peerj.19197_

## Round 0.1 · original submission · Major Revisions

All 3 reviewers have provided extensive and valuable comments. Please respond to them all in an appropriate revision

Note: Reviewer 3 has provided a PDF

Reviewer 1 ·

Basic reporting

While the authors report elevated ONSD values at 24 hours, they have not established a clear threshold to differentiate between ONSD and stroke prognosis. This omission limits the clinical applicability of their findings. It is essential to define a specific cut-off value that can be used to assess the relationship between ONSD and patient outcomes in stroke cases. Without this, the use of ONSD as a prognostic tool in stroke patients remains limited.

Experimental design

1.Measurement Timing of ONSD:
The authors have chosen to measure the ONSD values at a single time point of 24 hours. It is unclear whether measurements were taken at other time points. Could the authors provide insight into the temporal variation of ONSD values? Understanding how these values change over time is crucial for interpreting the findings.

2.Comparison Between Measurement Techniques:
The study employs CT to measure ONSD, while other research has utilized ultrasound for this purpose. It would be beneficial for the authors to discuss the differences between these two methodologies. Specifically, what are the advantages and disadvantages of using CT versus ultrasound in this context?

Validity of the findings

Patient Information:
The manuscript should include a comprehensive table summarizing patient information. This will enhance the reader's understanding of the study population and their characteristics.

Additional comments

Materials & Methods Section:
There is an instance of incorrect formatting regarding the institution’s name (XXX University) in the Materials & Methods section.

·

Basic reporting

The writing is generally clear. I have identified a few instances of potential ambiguity, that should be corrected.
In the introduction, there is extraneous detail unrelated to this paper, yet important factors, such as the strength of the evidence supporting the use of ONSD as a surrogate for ICP, and references for this, is absent.

Experimental design

This is original primary research falling within the aims and scope of the journal.
The explanation of how this research fills an existing knowledge gap could be better made (as above).
The experimental method is appropriate, however I do not believe that the most appropriate, or powerful, statistical analysis technique has been used, for the majority of the analyses (most should appropriately be analysed with paired T-tests measuring differences in the change in ONSD between dependent pairs (left and right eyes, affected, unaffected sides). This has the potential to change the main results.
I would ideally like to see more rigour around the measurement of ONSD, even a validation study in a subset, from an independent examiner.

Validity of the findings

As above - I have suggested a re-analysis of the data before considering a re-evaluation.

Additional comments

Analysis: I believe there is a more appropriate and powerful way to analyse the data that has the potential to change the primary findings. It would be more appropriate to also use a paired test to look at left versus right eyes, and affected versus unaffected sides, within individuals, as has been used to measure difference over time. These are clearly related (dependent) measures, and a paired test will significantly increase statistical power. i.e. my left ONSD is much more likely to be similar to my right, than it is to yours. For affected v unaffected side, I believe the relevant measurement is not the absolute measures but the change on affected v unaffected sides pre and post thrombolysis (a paired test measuring the difference in the change in diameter, between the two sides, i.e. is the affected side diameter more likely to enlarge more than the unaffected side?). Similarly for the survival test. It is the change in diameter over time, that one would expect to be a surrogate for ICP, and that is what should be compared between surviving and non-surviving patients? Possibly this may be in particular on the affected side (but I doubt you will show a difference – ICP tends to be relatively evenly distributed in the anterior compartment). So assuming on the re-analysis there is no difference, then would suggest the analysis should be of the mean change in diameter between sides, for each patient),
When reporting paired tests, you are measuring the difference between measures, so I think this is what should be reported, not the mean of the groups (it is more powerful, these are not just group differences, you measured a change). To give context, you can also report the mean. E.g. “ONSD increased from the prethombolytic, to the post-thrombolytic scan, by a mean of +0.2±0.xmm (p<0.001), from a mean baseline ONSD of 5.3±0.7mm.

Editorial comments
Introduction
There is extraneous detail that is not needed for the interpretation of this paper, and could be omitted. It is not a review paper, so detail such as the molecular pathophysiology of cytotoxic oedema is unnecessary to understanding why this research was undertaken, and is not needed/helpful. Moreover, ONSD is primarily related to raised ICP, not to cerebral oedema, so it is not clear why oedema is being discussed at all. These are not equivalent. For example, idiopathic intracranial hypertension, or hydrocephalus, may result in increased ICP without oedema. There is also some experimental and preliminary clinical evidence that in the first 24 h after ischaemic stroke, ICP may rise without measurable oedema, though clearly at later times (3-5 days) oedema is important.

Much more relevant than explanation of pathophysiology of oedema, would be a very brief summary of the data supporting the use of ONSD measurement as a surrogate for intracranial pressure. In particular, I would explain more about why repeat measurements of ONSD are likely to be much more powerful than a single cut off value for identifying ICP elevation (it negates difference due to normal variation in the population).
Because in most instances it will be apparent that it is a follow up scan, and the amount of oedema will be able to be estimated from the scan, there is a risk of inadvertent bias influencing measurements. Were any steps taken to try to minimise this risk? Was there any validation, to compare interrater agreement? If the images are still available, this would be quite helpful, even if only performed on a subset of the patients.

Minor editorial comments:
“It is predicted that patients with clinical improvement” should be changed to “we predicted that…” (all statements should be attributable to someone)

I would suggest dropping the final sentence of the introduction. It has an element of predicting the results. Better to finish with the aim.

Materials and methods:
“those who could not measure ONSD in both eyes” should be changed to “those in whom ONSD could not be measured in both eyes” (the patients are not doing the measuring)
“those who had previously been diagnosed with ICP” should read “elevated ICP” (everyone has an ICP)

All abbreviations should be defined on first use – e.g. CCT

Table 2 “Affected” not “Effected”

I have not reviewed the results, results tables, or discussion in much detail, since I think there is a reasonable chance that with re-analysis, the main results may change, and thus change these sections significantly.

Reviewer 3 ·

Basic reporting

Structure conforms to PeerJ standards, discipline norm, or improved for clarity. Additional comments are attached.

Experimental design

Rigorous investigation performed to a high technical & ethical standard.
Methods described with sufficient detail & information to replicate.

Validity of the findings

Meaningful replication encouraged where rationale & benefit to literature is clearly stated.

Annotated reviews are not available for download in order to protect the identity of reviewers who chose to remain anonymous.

---

## Round 0.2 · Major Revisions

Dear Author,

There remain numerous typographical errors and issues with scientific writing throughout the manuscript. Please consult a scientific writing expert to refine the language and improve the presentation of the results. Additionally, ensure that the title accurately reflects the design of the study.

Best regards.

Reviewer 1 ·

Basic reporting

I have carefully reviewed the author’s revised manuscript and responses, and I have no further questions.

Experimental design

I have no questions regarding this part.

Validity of the findings

I have no questions regarding this part.

·

Basic reporting

Modifications have addressed all my prior comments

Experimental design

No Comment

Validity of the findings

No Comment

Additional comments

Lines 134-135 – repeated use of “ONDS”. Should be changed to ONSD

Line 175 would be better phrased “Fourteen (15%) patients who received thrombolytics developed intracranial complications”

---

## Round 0.3 · accepted · Accept

The authors have satisfactorily replied to the reviewers and the article is ready for publication.